# Culturable Yeasts as Biofertilizers and Biopesticides for a Sustainable Agriculture: A Comprehensive Review

**DOI:** 10.3390/plants10050822

**Published:** 2021-04-21

**Authors:** María Hernández-Fernández, Gustavo Cordero-Bueso, Marina Ruiz-Muñoz, Jesús M. Cantoral

**Affiliations:** Laboratory of Microbiology, Department Biomedicine, Biotechnology and Public Health, University of Cádiz, Puerto Real, 11510 Cádiz, Spain; maria.hernandez@uca.es (M.H.-F.); marina.ruiz@uca.es (M.R.-M.); jesusmanuel.cantoral@uca.es (J.M.C.)

**Keywords:** yeasts, sustainability, biocontrol, growth promoters, bioinsecticides, antifungals, biostimulants

## Abstract

The extensive use of synthetic fertilizers and pesticides has negative consequences in terms of soil microbial biodiversity and environmental contamination. Faced with this growing concern, a proposed alternative agricultural method is the use of microorganisms as biofertilizers. Many works have been focused on bacteria, but the limited literature on yeasts and their potential ability to safely promote plant growth is gaining particular attention in recent years. Thus, the objective of this review is to highlight the application of yeasts as biological agents in different sectors of sustainable agricultural practices through direct or indirect mechanisms of action. Direct mechanisms include the ability of yeasts to provide soluble nutrients to plants, produce organic acids and phytohormones (indole-3-acetic acid). Indirect mechanisms involve the ability for yeasts to act as biocontrol agents through their high antifungal activity and lower insecticidal and herbicidal activity, and as soil bioremediating agents. They also act as protective agents against extreme environmental factors by activating defense mechanisms. It is evident that all the aspects that yeasts offer could be useful in the creation of quality biofertilizers and biopesticides. Hence, extensive research on yeasts could be promising and potentially provide an environmentally friendly solution to the increased crop production that will be required with a growing population.

## 1. Introduction

Fertilizers are essential to foster the growth of high yield crops. Of the basic nutrients that plants need for healthy growth, large amounts of nitrogen (taken up as NH_4_+ or NO_3_^−^), phosphorus (taken up as H_2_PO_4_^−^), calcium (taken up as Ca^2+^), sulfur (taken up as SO_4_^2−^), magnesium (taken up as Mg^2+^), potassium (taken up as K^+^), iron (taken up as Fe^2+^ or Fe^3+^) and zinc (taken up as Zn^2+^ or Zn(OH)_2_) are required by many crops on most soils [1]. Such large amounts of nitrogen, phosphorus, sulfur and potassium nutrients are supplied mainly in the form of mineral fertilizers, either processed natural minerals or manufactured chemicals [2]. The development and use of mineral fertilizers after the Second World War have provided significant increases in crop yield using less land, which assisted in supporting the burgeoning population [3].

Since the 1960s, there are increasingly studies with alarming data on animal and human health related to the widespread use of mineral fertilizers [3,4,5]. Likewise, the uncontrolled use of fertilizers has reduced microbial activity and soil fauna, causing a significant loss of biodiversity, as well as an increase in groundwater pollution, mostly from nitrates [6]. According to Regulation (EU) 2019/1009 of the European Parliament and of the Council on 5 June 2019, the different maximum amounts of nitrate to be added according to the fertilizer to be used are specified, as well as the rest of the compounds that compose it. 

Over the next thirty years, the world population is expected to reach almost 10 billion. Thus, in order to meet the estimated food demand in 2050, food and feed supplies will need to increase by 60% [7]. To avoid potential supply shortages, food production and distribution must be managed in a more efficient and sustainable way. In order to achieve this, some producers are turning to organic fertilizers which come from many different recycled sources, including manure, crop residues, and town sewage such as human waste [8]. Organic fertilizers offer some benefits that have contributed greatly to maintaining crop yields; however, they are usually lower in nutrients and less effective in supporting plant growth when used together with chemical fertilizers. For instance, the total nutrients in manure are less than 2% and the levels of phosphorus and nitrogen nutrients therein are more difficult to be effectively utilized. This is due to their losses into the environment, thus contributing to soil contamination and the eutrophication of lakes, underground aquifers and coastal areas [8]. Moreover, unprocessed organic fertilizers also affect animal and human health since they may contain pathogenic microorganisms, such as the harmful bacteria species *Escherichia coli*, or the genera *Campylobacter*, *Salmonella*, *Clostridium botulinum*, among others [9]. Furthermore, organic fertilizers can produce undesirable odors [10] and may contain toxic chemicals that can pose a risk to human, animal or plant health [11]. These and other harmful side effects have led the European Union and other countries to restrict the use of such fertilizers (European Commission Regulation, 2008).

Biostimulants are natural substances or microorganisms that are applied to plants with the aim of improving their nutritional efficiency, tolerance to abiotic stress and/or quality traits of crops, regardless of their nutrient content. Biofertilizers, a subcategory of biostimulants, can be defined as microbial inoculants, in which active or inactive formulations of beneficial microorganisms can improve the nutritional efficiency of plants [12,13,14]. Bhardwaj et al. (2014) reported that the use of biofertilizers raised crop yields around 10–40%, which increased the content of proteins, essential amino acids, vitamins, and nitrogen fixation [15]. Consequently, biofertilizers have been proposed as an alternative to mineral fertilizers. For example, nitrogen and/or sulfur fixing microorganisms including bacteria, such as *Azotobacter*, *Azospirillum* and *Rhizobium*, and fungi, such as *Aspergillus niger* and *A. tubingensis* have been utilized in biofertilizers [16,17,18,19]. Genetically modified bacterial strains have also been developed and tested as biological fertilizers [20]. The application of bacteria and fungi as biofertilizing agents has generated growing enthusiasm for being an eco-friendly alternative to synthetic fertilizers. However, their still low effectiveness in comparison to conventional fertilizers reduce their current application [21,22,23]. 

Similarly, increased crop production relies on disease and pest control management. In fact, pest management is the main aspect of obtaining high and healthy crop yields [24]. Approximately 50,000 species of fungi, 18,000 species of weeds and 10,000 species of insects have been described as damaging crops and food worldwide [25]. In order to control them, synthetic pesticides are the most widely used alternative. However, their inappropriate and extended use have led to resistance in these organisms. Furthermore, their non-biodegradability and toxicity have led to reconsideration of the use of sustainable pest control measures. In this regard, Annex II of Regulation (EC) 1107/2009 of the European Parliament and of the Council of 21 October 2009, concerning the commercialization of phytosanitary products, introduced so-called cut-off criteria, which directly prohibit the use of numerous substances, such as pesticides. Thus, the application of microbial pesticides appears as a promising alternative [26]. Biopesticides are products containing microorganisms that are widely used for pest control management in sustainable agriculture [27]. Nowadays, biopesticides are only used in 5% of crops. Therefore, the research and introduction of new microbial agents that can protect plants is a fundamental strategy to cover new needs in this field [28]. In this regard, there is a current growing interest in the study of yeasts with potential to be used as biofertilizers or biopesticides. 

Yeasts are single-cell fungi that can assimilate different carbon and nitrogen sources. Because they are facultatively anaerobic, some also have the ability to ferment carbohydrates. Its main mode of reproduction is asexual, by budding or cell fission. This characteristic allows them to adapt to growth both in liquid environments, whether in suspension or forming biofilms, as well as solids, including sand grains [29]. Due to their wide nutritional activity and their easy proliferation capacity, yeasts are relatively abundant in the edaphic ecosystem [30]. As a consequence of their adaptation to the habitat, they can assimilate a large number of toxic compounds from synthetic pesticides, thus contributing to soil bioremediation [31]. Considerable progress has been made in increasing research showing that plant growth can be enhanced by yeasts. Some of their positive effects could be due to the provision of soluble nutrients and the production of different phytohormones and enzymes that directly contribute to plant growth, acting as biostimulants [32,33,34]. 

Biological control is another mechanism in which antagonistic microorganisms can be useful. The modes of action usually involved include: (i) competition for nutrients and space; (ii) production of toxins; (iii) secreted enzymes; (iv) production of volatile organic compounds (VOCs); (v) parasitism; (vi) induced systemic resistance [12,35]. In this way, yeasts act as biocontrol agents through an antagonistic action against pathogenic microorganisms that indirectly affect plant growth, acting mainly as a biofungicide and, to a lesser extent, as a bioinsecticide or bioherbicide [36,37,38,39,40,41]. In response to extreme environmental factors such as salinity, high temperature, drought or metal toxicity affecting plants, several defense mechanisms are activated by yeasts, which lie in the production of compounds that reduce such stress [42,43]. Both the large variety of mechanisms used and the intrinsic characteristics they present make them a promising microorganism as an alternative in different fields of agriculture (Figure 1).

This review collects information from various studies carried out with yeasts to promote sustainable agriculture, working both as biostimulant and biopesticide. It also aims to highlight the need to strengthen the study of yeasts and their mechanisms of action. A better understanding of these new avenues of research could constitute a valuable role in the environment that would reduce and potentially eliminate the need for synthetic fertilizers.

## 2. European and International Regulations on the Use of Biostimulants and Biopesticides

Legislation on the use of biostimulants and biopesticides in different countries is not consistent, which has led to a biased variety and availability of fertilizers among farmers in different regions. To eliminate such biases, it is necessary to create a common marketplace for these substances. In addition, greater regulatory coherence accompanied by a better understanding of biostimulants and biopesticides would allow the reliability of their concept and could benefit the growth of their market [12,44]. 

### 2.1. Biostimulant Regulations

Microbial biostimulants are regulated at the national level, so the wide diversity of existing legislation has created some uncertainty among farmers [45]. However, the European Parliament has recently launched a new regulation, known as “Regulation European Union (EU) 2019/1009 laying down rules on the making available on the market of EU fertilizing products and amending Regulations European Commission (EC) 1609/2009 and EC 1107/2009”, which aims to harmonize the EU biostimulant market. They indicated that microbial plant biostimulant will be constituted by the microorganism or group of microorganisms mentioned in the component materials category (CMC 7): *Azotobacter* spp., *Rhizobium* spp., *Azospirillum* spp. and mycorrhizal fungi. However, this regulation will not come into effect until July 2022. 

On December 2018, the USA Agriculture Improvement Act of 2018 became law. Currently, definitive guidance is being generated by the Environmental Protection Agency (EPA) for the regulation of plants, including biostimulants. USA and Europe are the two regions where biostimulants are defined and regulated. However, in other regions, biostimulants are covered by national laws in different categories (organic fertilizers, biofertilizers, plant growth enhancers or plant enhancers). For example, in India biostimulants’ main categories are biofertilizers and organic fertilizers, according to the Fertilizer Control Order (FCO) of 1985 and in Brazil, Lei Ordinária No. 6.894/1980 classifies biostimulants as inoculants or biofertilizers.

### 2.2. Biopesticides Regulations

In the EU, biopesticides are regulated as plant protection products under Regulation (EC) 1107/2009 of the European Parliament and of the Council concerning the placing of plant protection products on the market. Thus, this regulation does not recognize biopesticides as a regulatory category. However, reference is made to microorganisms, pheromones and biological products as far as plant protection product is concerned. It is well known that fewer biopesticides are registered in EU than in the United States, India or Brazil, due to the lengthy and complex registration processes involved [44].

In the USA, the term biopesticide is regulated by the following federal agencies: (i) the Environmental Protection Agency (EPA); (ii) the United States Department of Agriculture, the Animal and Plant Health Inspection Service (USDAAPHIS); (iii) the Food and Drug Administration (FDA). In India, the legislation for synthetic pesticides is the same as for biopesticides and is governed under the Insecticides Act (1968). The Central Insecticides Board (CIB) and the Registration Committee (RC) work under this Act as powerful bodies to regulate biopesticides [46]. In Brazil, the Decree 6.913 of 23 July 2009 referred for the first time to the term biopesticide as a phytosanitary product. The registration of biopesticides in this country is regulated through (i) the Ministry of Agriculture; (ii) the Brazilian Institute of Environment and Renewable Natural Resources (IBAMA); (iii) the National Health Surveillance Agency (ANVISA) [47].

## 3. Yeasts as Biostimulant and Biofertilizer Agents

Yeasts are mainly known worldwide for their application in the food industry, but their implementation in different markets has increased in recent years. Nowadays, yeasts are one of the model organisms for biomedical research [48,49] and they have had a great impact on the biofuel production industry [50,51]. Similarly, the agricultural sector has also benefited from their use, where research has grown recently [52].

Plant growth-promoting microorganisms (PGPMs) have several beneficial effects on plants due to both direct and indirect mechanisms. Compared to the use of bacteria and mycorrhizal fungi, the use of yeasts as PGPMs has not been extensively investigated. The largest yeast populations are generally found in the rhizosphere rather than in the soil [53]. In fact, a growing number of studies indicate that plant root growth may be favored by yeasts in the rhizosphere [43,54,55], although a wide diversity of soil yeasts has also been investigated for their potential as biofertilizers [32,56,57]. Alternatively, endophytic yeasts, which colonize the interior of the plant, can stimulate crop growth [58,59,60] as well as there are studies reporting epiphytic yeasts as potential plant growth promoters [60,61]. Their successful results highlight the importance of maintaining the biodiversity of the existing microbiota population in the soil and within plants, which is often diminished by the uncontrolled use of fertilizers. In addition, it should be noted that yeast suspensions in their foliar application could also play an essential role in plants [62,63].

Low availability of some essential nutrients in soluble form can stunt plant growth. Direct mechanisms, including molecular nitrogen (N_2_) fixation, phosphorus (P) and potassium (K) solubilization, solve this problem [64]. Other direct mechanisms are the production of organic acids and plant growth hormones such as auxins, cytokinins or gibberellins. Most attention is given to indole-3-acetic acid (IAA), which is the most common plant auxin responsible for regulating various physiological and growth aspects of plants [65]. As for indirect mechanisms, microbes enhance plant growth through components that induce resistance to environmental stress, leading to the production of antifungal compounds [66]. 

Some of the mechanisms described and discussed below have not been fully tested by molecular analysis yet, but they have been proposed based on similarities with other biological systems. However, the increasing number of annotated yeast genomes as well as the availability of different transformation techniques should make it possible to better understand different mechanisms and thus will unequivocally highlight the capability of yeasts as PGPMs in the near future.

### 3.1. Direct Plant Growth Mechanisms

#### 3.1.1. Nutrient Supply

It is known that one of the main causes of limited yield of most of lower quality crops worldwide is due to the inadequate use of nitrogen [67]. Despite the abundance of nitrogen present in the atmosphere, it is not available in a form suitable for being used by plants, which can only assimilate nitrite, nitrate or ammonia [66]. When using conventional nitrogen fertilization in agriculture, it is often lost during rainfall or by mineral leaching of these fertilizers. On the other hand, biological nitrogen fixation provides ammonium through the action of microorganisms, which in turn is transferred to the plant [66]. While bacteria are the main N_2_-fixing microorganisms, few is still known about this potential capability of yeasts. Despite this, it has been demonstrated that a strain of *Candida tropicalis* isolated from soil shows an interesting and great ability to fix nitrogen [68]. On the other hand, studies of ammonia-producing yeasts have been reported with the genus *Meyerozyma* as the major NH_3_ producer in the same way that *Pseudozyma rugulosa*, *Cryptococcus flavus* and *Pseudozyma antarctica* are [60,69]. In this way, it is important to highlight the enzyme 1-aminocyclopropane-1-carboxlyate (ACC), which is responsible for the cleavage of the ethylene precursor ACC into α-ketobutyrate and ammonia. Thus, the use of this enzyme has a double positive effect: the reduction of the amount of ethylene under adverse conditions, since ethylene can become detrimental to the plant in large quantities, and the release of ammonia, which can generate a nitrogen recycling mechanism for plants through the use of its symbiotic partners [60,70]. Yeasts such as *C. tropicalis* and *Cryptococcus* sp. have been described with ACC deaminase activity [32,71]. Furthermore, some yeasts have been reported as capable of performing the denitrification process [72]. This is an anaerobic respiration process in which nitrate or nitrite is the final electron acceptor, producing nitrous oxide (a greenhouse gas) as final product, contributing in turn to reduce eutrophication [73].

Phosphorus (P) is the second most important key plant nutrient after nitrogen [74]. Thus, P deficiency is also considered as one of the major limiting factors in crop productivity, especially in the tropic and subtropic regions. This is due to it is the least available element to plants in most soils, unlike other macronutrients [75]. However, soil microorganisms have generally been found to be most effective in making P available to plants from inorganic and organic sources by solubilization and mineralizing complex P compounds [76]. Among the various strategies adopted by microbes, the mediation of organic acids has made the most sense as the main means of inorganic P solubilization [74]. This is reported by the reduction of pH under in vitro conditions, which is largely due to the production of carboxylic acids by the microorganism [77]. The yeast *C. tropicalis* was able to solubilize Ca_3_(PO_4_) in the same way as *Rhodotorula* sp. and *Lachancea thermotolerans* providing soluble inorganic P with the consequent pH reduction [68,69,77]. Moreover, El-Latif and Mohamed (2011) demonstrated the ability of the some soil strains belonging to *Yarrowia lipolytica* and *Saccharomyces cerevisiae* to solubilize inorganic phosphate compounds by producing citric acid [78].

Potassium (K) is the third essential macronutrient for plants. It plays a fundamental role in several processes, including plant growth. The high amounts of K present in soils correspond to insoluble forms of silicate minerals, so K solubilization by microorganisms is also the main mechanism by which this element can be available to plants [79]. Some soil microorganisms are capable of releasing K from minerals by excreting organic acids that can enhance dissolution by a proton or ligand-mediated mechanism, thus promoting a reduction in the pH [80]. Results of a study on potassium solubilization from ultramafic alkaline rock dust by the yeast *Torulaspora globosa* showed high capacity to solubilize K minerals, being acid production the main mechanism used by the yeast [54]. In another experiment, *Rhodotorula glutinis* and *Pichia anomala* recorded the highest concentration of K released from mica with consequent significant reduction in pH. In addition, it was shown to promote plant height, root and shoot dry weights [80].

PGPMs inoculation can also increase plant uptake of several other nutrients such as calcium, iron, magnesium, sulfur and zinc. This uptake generally occurs during acidification of the soil rhizosphere through organic acid production. While studies have been found reporting Ca and Mg solubilization and S oxidation activity by bacteria [81,82,83], hardly any have been reported in yeasts. Falih and Wainwright (1995) reported the yeasts *Williopsis californica* and *S. cerevisiae* as capable of oxidizing elemental S in vitro to produce thiosulfate, tetrathionate and sulfate [84], but no yeasts with this ability have been reported recently. The same cannot be said for iron and zinc, as there are multiple studies of how yeasts can facilitate the assimilation of these nutrients [43,68,69]. Regarding the importance of Zn in plant nutrition, it should be noted that its deficiency causes the development of anomalies in plants, such as stunted growth and smaller leaves, thus highlighting the need to control this nutrient [85]. As for Fe uptake, plants can release phyto-siderophores. However, these tend to have a lower affinity for iron compared to microbial siderophores [86]. Therefore, PGPMs can chelate Fe^3+^ to make it available to plant roots, absorbing it through the degradation of chelating agents and their consequent release of iron, the direct uptake of siderophore-Fe complexes or through a ligand exchange reaction [87]. 

#### 3.1.2. Phytohormones

Yeasts can produce some plant growth-promoting phytohormones. Among these regulators are auxins, which are a group of compounds that contain an indole ring [65]. Indole-3-acetic acid (IAA) is the major plant growth promoter member of the auxin class which is known to stimulate rapid and long-term responses in plants by regulating various physiological and developmental processes [88]. Studies on IAA-producing yeasts, including *Rhodosporidium paludigenum*, *S. cerevisiae*, *Aureobasidium pullulans*, *Candida* sp., *Dothideomycetes* sp, *Hanseniaspora uvarum*, *Meyerozyma caribbica*, *Meyerozyma guilliermondii*, *Torulaspora* sp., *Barnettozyma californica*, *Cryptococcus laurentii*, *Rhodosporidiobolus fluvialis*, *Candida maltosa* and *Pichia kudriavzevii* among others, have been reported [43,65,68,69,89,90,91,92]. While auxin biosynthesis has not yet been fully elucidated, remarkable progress has been made over the last few years in understanding the mechanism of IAA biosynthesis in yeasts. Two main pathways have been proposed: tryptophan(Trp)-dependent and Trp-independent pathways [89]. As for the Trp-dependent pathway, the possible routes to be used are indole-3-pyruvic acid (IPA), tryptamine (TAM) and indole-3-acetamide (IAM) [89]. It has been demonstrated that the IPA pathway is the main route for the biosynthesis of IAA in yeasts supported by studies with *S. cerevisiae* [92], *R. paludigenum* [65] and *R. fluvialis* [89]. The first step is the conversion of Trp to IPA by aminotransferase activity, then IPA is carboxylated to indole-3-acetaldehyde (IAAld) by indole-3-pyruvate decarboxylase (IPDC) activity and the last step is an oxidation of IAAld to IAA. Alternatively, IPA can also be converted directly to IAA by indole-3-pyruvate monooxygenase [89]. Because IAA production is regulated by several factors such as the strain used, its growth phase, the precursor concentration, or medium components, each yeast has a characteristic biosynthesis pathway [93]. Regarding the Trp-independent pathway, it has been shown that *S. cerevisiae* is able to produce IAA under these conditions; however, the intermediate steps and genes involved in this pathway remain undefined [92]. In addition, Fernandez-San Millan et al. (2020) reported that 97% of the yeasts used in the study showed Trp-independent IAA biosynthesis capacity [69]. Nevertheless, they suggested that the Trp-dependent pathway was more effective since a higher amount of IAA was obtained.

Another group of phytohormones are the cytokinins, organic molecules that promote plant growth through facilitated cell division and growth. Zeatin is one of the most common cytokinins, which has been reported to be synthetized by some yeasts such as *Sporobolomyces roseus*, *Metschnikowia pulcherrima* and *A. pullulans*. As with auxins, zeatin synthesis in culture varies among species and strains, and may depend on culture conditions [94]. Additionally, gibberellins, commonly known as gibberellic acids, are another important plant growth regulator. Among their abilities, they can stimulate seed germination or trigger the juvenile stage to the adult leaf [95]. The literature does not provide many studies of gibberellic acid production by yeasts. However, a study recently resulted in a high production of gibberellic acid by some of the yeasts isolated from soil, highlighting its potential as a plant growth promoter [96].

### 3.2. Indirect Plant Growth Mechanisms

As discussed above, plants are exposed to a continuous combination pathogenic attacks and extreme environmental factors that limit crop yields. In fact, the latter leads to more than 50% of crop reductions, which could be increased due to climate change [97]. These include extreme levels of light, ultraviolet radiation, temperature, drought, salinity or metal toxicity [98]. To counteract these stresses, plants are equipped with a large array of defense mechanisms. The accumulation of some functional substances, such as polyamines, induces resistance against extreme environmental stresses [99]. Moreover, they are also involved in various processes of plant growth and development. Thus, polyamines (e.g., spermidine, spermine, putrecine), which consists in low molecular weight aliphatic amines, are considered a new type of plant biostimulant [100]. One of the most common plant hormones that mediates the response to stressors is ethylene [101]. However, when ethylene is produced above its threshold level, it is detrimental to the plant [102]. As discussed above, the enzyme ACC deaminase cleaves ACC to form α-ketobutyrate and ammonia. Since ethylene is synthesized from ACC, its depletion affects the biosynthesis of the stress hormone ethylene in host plants and stimulates plant growth [102]. Several studies aimed at evaluating the plant-promoting effect of yeasts tested polyamine and ACC deaminase production with promising results. In this sense, polyamine-producing activity and ACC deaminase activity was reported for the yeast *C. tropicalis* [32]. A large number of yeasts, including *P. rugulosa*, *P. antarctica*, *A. pullulans* and *Dothideomycetes* spp. are polyamine-producing yeasts [43,60]. However, few yeasts have been reported to show ACC deaminase activity among which includes *Cryptococcus* sp. [71].

## 4. Yeasts as Biocontrol against Fungal Pathogenic Microorganisms

Biological control of fungal attack is also important to consider. As mentioned above, plant growth can also be indirectly influenced by the production of antifungal compounds. Many studies have shown that some yeasts could represent an alternative against fungal infections. It is well known that fungi are proficient in colonization and competition for space and nutrients than other groups of microbes that exist on the surface of fruits and vegetables [103]. Furthermore, yeasts do not usually produce toxic secondary metabolites such as antibiotics, allergenic spores or mycotoxins in their inhibitory activity [37,104], so they can be used as an effective biocontrol agent. 

### 4.1. Competition for Nutrients and Space

Since yeasts have different antagonistic properties depending on different factors, including the pathogen, the host and the environment, it is essential to better known about their modes of action in order to improve their viability and increase their potential in disease control. Competition for nutrients and space is considered the main mode of action by yeasts applied post-harvest, especially in fruits [105]. In terms of space, yeasts usually have the advantage of rapid and high growth leading to the formation of biofilms covering the entire wound area fruit [106]. In addition, competition of yeasts for nutrients against some pathogens is also successfully demonstrated in different studies, showing a rapidly depleting of nutrients, thus preventing the germination of fungal spores. In this sense, the ascomycete *A. pullulans* showed a great biocontrol activity against *Penicillium expansum* through competition for nutrients [107] and against *Monilinia laxa*, both through competition for nitrogen compounds and for space [108]. Moreover, iron plays an important role in the growth and virulence of pathogens. Its sequestration, either by pulcherrimine produced by *M. pulcherrima* [38,109] or by siderophores produced by other yeasts such as *R. glutinis* [110], hinders the germination and growth of pathogenic fungi due to the lack of iron [36]. 

### 4.2. VOCs and Killer Toxins Production

On the other hand, yeasts can also produce VOCs, which play an important role in pathogen control [111]. They could be considered as ideal antimicrobials since no physical interaction between the biocontrol agent and the food or pathogen is necessary. Antagonistic yeasts of the genus *Hanseniaspora* [112] and *A. pullulans* [113] showed inhibition efficacy against *B. cinerea* by producing VOCs. Similarly, killer toxins can also control post-harvest pathogens [35]. Yeasts that produce these toxins are usually resistant to them and to those of the same class, while they are lethal to other yeast strains [114]. Since the cell wall is one of the most common sites of action of toxins, different components can be found acting as receptors of killer toxins, such as β-1,3-D-glucans, β-1,6-D-glucans, mannoproteins and chitin [114]. Some known toxins produced by yeasts are: (i) PMKT and PMKT2, produced by *Pichia membranifaciens*, which bind, respectively, to β-1,6-D-glucans and mannoproteins of the cell wall of pathogens [115]; (ii) panomycocin, produced by a strain of *P. anomala*, which exerts its activity by hydrolyzing β-1,3-glucans [116]; (iii) zymocin, produced by *Kluyveromyces lactis*, which hydrolyzes chitins present in the fungal cell wall acting as antifungal components [117]. It should be noted that only a small fraction of the recognized killer toxins has been so far characterized in detail [105].

### 4.3. Parasitism and iNduced Systemic Resistance

Parasitism is another mechanism of action whereby there is a direct interaction between the antagonist and the pathogen. The secretion of lytic enzymes and the active development of the antagonist on the pathogenic fungus depends directly on the contact and recognition between them, which usually results in the death of these cells. There is evidence that *Candida famata*, *Rhodotorula mucilaginosa*, *Wickerhamomyces anomalus* and *P. guilliermondii* use parasitism for the biocontrol of *Colletotrichum gloeosporioides* by secreting enzymes such as β-1,3-glucanase [118,119]. In the same way, induced systemic resistance, mediated by beneficial microbes, is particularly important in biocontrol yeasts. Their responses are effective against necrotrophs [120] and are mediated by the activation of latent defense mechanisms, enhancing the activities of defense-related enzymes [121,122]. It has been possible to demonstrate the beneficial effects of induced resistance in the postharvest environment by treatments with the yeasts *R. paludigenum* and *W. anomalus.* They significantly reduced the incidence of *Penicillium* fungal disease through the production of different enzymes like β-1,3-glucanase [123,124]. Therefore, cell wall degrading enzymes are regaining importance through multiple mechanisms of action. Their study has been made possible by the development of high-throughput sequencing technologies that allows to study changes in gene expression in both host and antagonist tissues, providing information about the origin and regulation of these enzymes [37].

## 5. Yeasts as Bioinsecticide, Bioherbicide and Biodegrader

Biological control also includes insect attacks on plants or the production of weeds that hinder plant growth under ambient conditions. The indiscriminate use of pesticides has led during the last years to the accumulation of toxic residues in food, soil, air or water, as well as the growth of resistance in pests. The development of microbial strategies for the elimination of different substances in a sustainable way is being considered in recent years [125]. It was reported for the first time that the yeast *R. mucilaginosa* could degrade the neonicotinoid insecticides acetamiprid (AAP) and thiacloprid (THI) both in crop and soil, thus promoting bioremediation [31]. Another study addresses the esterase-mediated biodegradation of the insecticide chlorpyrifos by the yeasts *R. glutinis* and *Rhodotorula rubra* [126].

In recent years, there have a growing number of studies in which some yeasts’ capabilities to cope with insect pests have been proposed. RNA interference (RNAi) has made important advances as a developmental tool for pest management. It is based on production of double-stranded RNA (dsRNA) in plants and its subsequent ingestion by insects. Down-regulation of genes targeted by the ingested dsRNA via RNAi can trigger reduced growth or the pest’s death [127]. In this context, a genetically modified yeast has been able to express dsRNA targeting *y-tubulin* from *Drosophila suzukii*. Their results showed that this specific biopesticide decreased larval survival and that the sequence-dependent deletion of the target gene via RNAi was the main cause of the decreased fitness observed in *D. suzukii* [40]. Additionally, a combination of *Cydia pomonella* granulovirus and yeasts such as *M. pulcherrima*, *Cryptococcus tephrensis* or *A. pullulans* increased larval mortality and improved fruit protection against apple moth larval infestation. Yeasts stimulate larval feeding as the larvae move to ingest the virus [128]. Another promising approach is the use of food-associated microbial VOCs to trap insect pests in crops. *D. suzukii* seems to respond to olfactory signals produced mainly by *H. uvarum* in field experiments, which could be translated into an opportunity for the development of an attract-and-kill control technique [129,130,131]. Similarly, pheromone-based methods for pest management are frequently used. Due to the high cost of pheromone chemical synthesis production, alternatives are needed. In response, recent studies have shown that pheromone components or precursors can be manufactured from engineered yeasts. The fermentation of the engineered yeast *Y. lipolytica* produced the sex pheromone of the *Helicoverpa armigera* worm, which was successfully efficient in field experiments [132].

Herbicides are an important component in food safety. However, the lack of feasible sustainable alternatives requires the constant use of synthetic herbicides. In addition, the increasing demand for food production is leading to a growing market for synthetic herbicides, resulting in high environmental pollution. In recent years, the development of microbial strategies for the elimination of the substances in a sustainable way is being considered [125]. There are many glyphosate-based herbicides [133], whose excess can be harmful to animals [134]. In this sense, the yeast *Solicoccozyma terricola* has the ability to degrade glyphosate in soil and is used as a nutrient source [135]. Besides, a strain of the yeast *Clavispora lusitaniae* was able to degrade pendimethalin, which is a micro-toxic dinitroaniline herbicide [136], and a strain of *P. kudriavzevii* managed to degrade atrazine, a member of the s-triazine group of herbicides [137]. This bioremediation capacity that yeasts often exhibited can also be considered a highly interesting approach in sustainable agriculture. While it is known that there are no feasible alternatives to synthetic herbicides, the literature reports that there are auxin herbicides, which are classified as auxins if they induce physiological and phenotypic effects similar to those induced by IAA [41,138]. Due to the large number of phytohormone-producing yeasts mentioned above, these could also be used as a great, yet unknown, alternative.

These different techniques proposed by the studies reviewed, in which yeasts play an important role through direct and indirect mechanisms, are quite promising (Table 1). However, a better understanding of the mechanisms of these microbial agents would be necessary in order to improve our knowledge about their possible effects on both the environment and human health.

## 6. Application of Yeasts in Commercial Biofertilizers and Biopesticides

The potential advantages of biofertilizers and biopesticides include: (i) a renewable source of nutrients; (ii) maintenance of soil health and microbiota; (iii) reduction of synthetic fertilizers by 25–30%; (iv) decomposition of plant residues; (v) safety for the environment; (vi) lack of resistance by pests, thus being an environmentally friendly, non-polluting and cost-effective method. However, there are also some limitations, including: (i) lack of availability of suitable and effective strains; (ii) difficulty in marketing as the product contains live organisms; (iii) limited period of activity; (iv) lack of awareness of farmers and inexperienced personnel; (v) low potency and high cost of production [148].

As it can be guessed from what was discussed above, there has been an increase in interest and opportunities in the development and commercialization of bioalternatives that contribute to food security and environmental concerns. Among the commercialized products, it should be highlighted BioGro 2, a multi-strain biofertilizer for rice, which has resulted in shinier, less disease-prone, cleaner and stronger stalks and higher grain yields. It consists in a strain cocktail of bacteria and the yeast *C. tropicalis* [149]. This yeast strain was initially selected for its ability to solubilize insoluble PO_4_ in agar medium, but it has also been shown to have a set of properties that promote rice plants’ growth [32]. The application of this biofertilizer is effective in reducing the use of conventional N fertilizers by up to 52% on farms without decreasing rice yields [150]. On the other hand, there is NutriSmart^®^, a biofertilizer composed of a group of *S. cerevisiae* yeast strains that fix atmospheric nitrogen, break down phosphate rock into soluble phosphate and enhance the amount of exchangeable potassium in the soil, so it is free of toxic components. Due to the unique characteristics of yeasts to decompose complex organic matter, various wastes including animal manure, compost and sludge, are often used as substrates. Once applied to the soil, yeasts decompose the residues and release essential elements such as N, P and K. Thus, crop residues can be used to produce these biofertilizers by entering a sustainable recycling loop. Moreover, it has been demonstrated that it increases soil microbial activity while allowing to reduce the use of synthetic fertilizers by 20–30% [151].

Commercial biofungicides include Zimevit, which combines *Bacillus subtilis strain* UYBC38 and *M. pulcherrima* strain M26. It has been tested for the control of grapevine gray mold caused by *B. cinerea* with considerable success [152]. Other commercial antifungal formulations currently traded using yeasts include: Noli or Shemer ™, based on the yeast *M. fructicola* strain NRRL Y-27328 and commercialized in the Netherlands. Their target pathogens are phytopathogenic fungi belonging to the genera *Botrytis*, *Penicillium*, *Rhizopus* and *Aspergillus* [39,153,154,155,156]; Nexy, formulated with *Candida oleophila* strain O and marketed in Belgium. This product is effective against *Botrytis* and *Penicillium* [157,158,159]; Botector and BoniProtect™, marketed in Austria, is developed with two antagonistic strains of *A. pullulans*. The target pathogens in this case are *Penicillium*, *Botrytis* and *Monilinia* (DSM 14940 and DSM 14941), for what it is commonly used for pre-harvest control [160,161,162,163]. More biofungicides have been developed with yeast formulations but are not currently being used: Candifruit ™, supported by the yeast *Candida sake* CPA-1 and produced in Spain, effective against *Penicillium*, *Botrytis* and *Rhizopus* [164,165,166]; YieldPlus, based on *Cryptococcus albidus* and developed in South Africa. Their target pathogens were *Botrytis*, *Penicillium*, and *Mucor* [167,168]; Aspire, formulated with *C. oleophila* strain I-182 and marketed in the USA. It was effective against *Botrytis*, *Penicillium* and *Monilinia* [169,170].

Biofertilizer and biopesticide commercialization remains low globally but is steadily expanding. So much so that no commercial insecticide or herbicide containing yeasts seems to exist so far. In the developed world, where agricultural chemicals remain relatively inexpensive, the use of PGPMs occupies a smaller, but also growing, group [171]. Recent research has revealed yeasts as a promising microorganism for agriculture and has led to improved techniques that can reduce the impact on biodiversity. In fact, the patent literature testifies the industrial interest yeasts application for the design of tailored biotechnological solutions [172]. For example, recent patents proposed specific *Metschnikowia fructicola* strains to improve plant performance [173] and to inhibit the growth of unwanted microorganisms [174,175]. However, as discussed above, the European Union (EU) Regulation 2019/1009 does not yet consider yeasts as a possible microorganism to be added to biostimulants, so a more widespread use of these will require appropriate regulatory and legal frameworks that are currently strict and may hinder their proper use. Although they are a promising source that offers an alternative for sustainable agriculture, and they are still underexplored as a natural resource. Moreover, their biotechnological potential has not yet been fully explored and requires extensive research for industrial-scale development as a future perspective [176]. Important phases in this research could include: (i) search and development of competent microbial strains that have better host colonization potential and excellent field success; (ii) development of biofertilizers from multi-strain consortia to achieve effective field functions; (iii) the possibility of adding biological substances to improve product formulation; (iv) field experiments at “farmers’ scale” [177]. In addition, the production of VOCs or pheromones by yeasts can be proposed as a research challenge to develop a future bioinsecticide. In any case, yeasts would present plant growth promotion, biocontrol and pesticide remediation properties, contributing to boosting agricultural productivity.

## 7. Conclusions

The considerable growth of the world population in the coming years will increase the demand and need for agricultural strategies, including the use of conventional fertilizers. This expansion, in addition to causing the production and accumulation of residues, will also continue to cause the reduction of organic matter in soils. The modification of their physicochemical properties affects the levels of the microbial population, making it essential to search for new options that allow sustainable agriculture. As an alternative to the use of synthetic fertilizers, this review proposes yeasts as a PGPM capable of increasing agricultural productivity directly and indirectly, as well as contributing to soil bioremediation. The intrinsic characteristics of yeasts make them a promising microorganism in agriculture practices because of their usage facilities. Thus, the exploitation of its potential may be a promising technique, although it is necessary to better known the mechanisms of action by which PGPMs achieve the benefits in plants and its behaviors in field experiments. The discovery and development of efficient yeasts inoculant consortia that can act under diverse conditions with various plant species, their subsequent optimization, and industrial bioformulation might be one of the bases for developing future biofertilizers. These will be cost-effective, economical and socially acceptable as they will contribute to environmental friendliness and ensure a secure food supply, thus potentially providing a solution.

## Figures and Tables

**Figure 1 plants-10-00822-f001:**
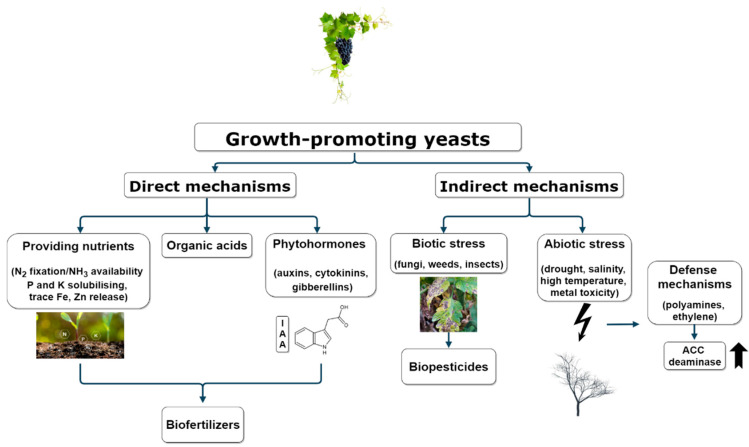
Modes of action of yeasts promoting plant growth.

**Table 1 plants-10-00822-t001:** Applications of yeasts as plant growth promoters, biofungicides, bioinsecticides or biodegraders.

Mechanism	Attribute	Yeasts	Reference
N_2_ fixation	Plant growth promotion	*C. tropicalis*	[68]
NH_3_ production	Plant growth promotion	*M. guilliermondii, P. rugulosa, C. flavus, P. antarctica,**Meyerozyma* sp., *M. caribbica*	[43,60,69]
P solubilization	Plant growth promotion	*C. tropicalis*, *L. thermotolerans*, *Rhodotorula* sp., *H. uvarum*, *Y. lipolytica, S. cerevisiae*	[32,68,69,77,78]
K solubilization	Plant growth promotion	*T. globosa*, *R. glutinis*, *P. anomala*	[54,80]
S oxidizing	Plant growth promotion	*W. californica*, *S. cerevisiae*	[84]
Zn solubilization	Plant growth promotion	*C. tropicalis*, *L. thermotolerans*, *Dothideomycetes sp.*	[43,68,69]
Polyamine production	Plant growth promotion	*C. tropicalis*, *P. rugulosa*, *P. antarctica*, *A. pullulans*, *Dothideomycetes* sp.	[32,43,60]
ACC deaminase activity	Plant growth promotion	*C. tropicalis*, *Cryptococcus* sp.	[32,71]
IAA production	Plant growth promotion	*C. tropicalis*, *M. guilliermondii, S. cerevisiae, A. pullulans, H. uvarum,**M. caribbica*, *W. californica*, *C. laurentii*, *R. fluvialis*, *C. maltosa*, *P. kudriavzevii*, *R. paludigenum*	[43,65,68,69,89,90,91,92]
Cytokinin production	Plant growth promotion	*A. pullulans*, *S. roseus*, *M. pulcherrima*	[94]
Pulcherrimine production	Biofungicide	*M. pulcherrima*	[38,109]
Siderophore production	Biofungicide	*R. glutinis* against *B. cinerea* and *P. expansum*	[110]
Competition for nutrients and space	Biofungicide	*A. pullulans* against *P. expansum* and *M. laxa*; *Issatchenkia terricola*, *P. anomala*, *M. pulcherrima*, *S. cerevisiae*, *Schizosaccharomyces pombe*, *H. uvarum* and *P. kluyveri* against *B. cinerea*	[38,107,108,139,140,141,142,143]
Parasitism	Biofungicide	*M. guilliermondii*, *P. anomala*, *C. famata* and *R. mucilaginosa* against *C. gloeosporioides*; *P. anomala*, *A. pullulans* and *P. kluyveri* against *B. cinerea*	[38,118,119,140,143]
VOCs production	Biofungicide	*A. pullulans*, *Hanseniaspora* sp., *P. anomala*, *M. pulcherrima*, *P. membranifaciens*, *M. guilliermondii*, *H. uvarum*, *Starmerella bacillaris*, *Candida pyralidae* and *P. kluyveri* against *B. cinerea*	[38,112,113,140,143,144,145,146,147]
Killer activity	Biofungicide	*P. membranifaciens*, *P. anomala*, *K. lactis*	[115,116,117]
Induced systemic resistance	Biofungicide	*R. paludigenum* against *P. digitatum*; *P. anomala* against *P. expansum*	[123,124]
Insecticide biodegradation	Biodegrader	*R. glutinis* and *R. rubra* degrades chlorpyrifos; *R. mucilaginosa* degrades neonicotinoid insecticide; *S. terricola* degrades glyphosate insecticide	[31,126,135]
Herbicide biodegradation	Biodegrader	*C. lusitaniae* degrades dinitroaniline herbicide; *P. kudriavzevii* degrades s-triazine group herbicides	[136,137]
Insecticidal activity	Insecticide	*Y. lipolytica*, *A. pullulans*, *H. uvarum*, *M. pulcherrima*, *C. tephrensis*	[128,129,131,132]

## Data Availability

Not applicable.

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
