# Peer review of "Culturable Yeasts as Biofertilizers and Biopesticides for a Sustainable Agriculture: A Comprehensive Review"

_plants, 2021, doi:10.3390/plants10050822_

Round 1

Reviewer 1 Report

This manuscript presents a review on the application of yeasts for plant growth promotion and pre- and post-harvest biological control of plant pests and diseases. Microorganisms, including yeasts, can potentially replace chemical fertilizers and pesticides and there is currently big interest in the use of such biological methods for obtaining a more sustainable agriculture. The manuscript adopts a valuable approach by treating yeasts for both biostimulating and biocontrol (and to a lesser degree biodegradation) purposes in the same review, which is not so often done. Indeed, similar organisms/yeasts can be useful for the different intended effects and sometimes it can be difficult to know exactly which is the dominant mechanism for a beneficial effect.    

However, unfortunately this manuscript has several considerable weaknesses that have to be taken into consideration:

  • In my opinion, the scientific reasoning and argumentation is shallow in several places and a few examples are given in the detailed comments below.
  • Although I consider it an interesting move to treat both growth promotion and pest control by yeasts application in the same study, it is puzzling that the former is given much more attention. For instance, the current summary in the manuscript of regulatory conditions for biostimulation products seems useful. But its inclusion appears inconsistent, since nothing is said about the fact that microorganisms for pest control are in most countries regulated within pesticide frameworks, as ‘biopesticides’ or ‘microbial pesticides’. Also, there is comparatively little information on microbiological pest/disease control in the introduction section.
  • The English in the manuscript needs substantial improvement. The text also contains quite a few repetitions and to help the reader it needs better organization, possibly with shorter paragraphs and enhanced use of sub-headings.

A few specific comments and suggestions:

L 77-80: Unclear sentence. Yeasts may not have the same effectiveness as chemical fertilizers but of course they might have other benefits. And it is not clear which ‘problem’ is referred to here.

L 83-84: This sentence almost completely lacks understandable and useful content. It does not provide a good definition of yeasts, and the statement about assimilation is principally valid for all microorganisms.

L 167-169: This sentence is extremely vague. Does it imply that since the positive effects are ‘preserved’, the microorganism needs to be applied only once? And how can something be preserved under natural conditions in in vitro conditions?

L 198: It is not clear what is meant by ‘the cycle of sustainable agriculture’.

L 212: Lachancea is misspelled.

L 381-382: In my view the statement ‘encouraging results that consider yeast as a natural insecticide’ is scientifically problematic. First, ‘results’ cannot ‘consider’ yeasts as something. Second, the word ‘natural’ is vague and can be highly (positively) value-laden, and in my opinion should be very cautiously used in scientific texts.

L 422-423: It is difficult to understand what the authors want to say here. Why would a better mechanistic understanding of the mechanisms of plant beneficial yeasts increase safety for consumers? And it is difficult to see a direct connection between measures to ensure a more sustainable agriculture and the safety of plant produce for human consumption.

Table 1: The table would benefit from better organization, perhaps by grouping studies according to type of effect and/or the mechanisms.

L 448-474: An updated overview of commercially available organisms and products is welcome. I am not sure, however, that all these products are still available on any market. Additionally, information about in which countries the products are authorized would also be valuable.

L 506-507: This sentence lacks scientific depth. What is meant by a yeast being ‘stable’? And what could possibly be the ‘climatic basis’ for any kind of ‘stability’ in a yeast?    

Author Response

Comments to reviewer 1:

This manuscript presents a review on the application of yeasts for plant growth promotion and pre- and post-harvest biological control of plant pests and diseases. Microorganisms, including yeasts, can potentially replace chemical fertilizers and pesticides and there is currently big interest in the use of such biological methods for obtaining a more sustainable agriculture. The manuscript adopts a valuable approach by treating yeasts for both biostimulating and biocontrol (and to a lesser degree biodegradation) purposes in the same review, which is not so often done. Indeed, similar organisms/yeasts can be useful for the different intended effects and sometimes it can be difficult to know exactly which is the dominant mechanism for a beneficial effect.    

However, unfortunately this manuscript has several considerable weaknesses that have to be taken into consideration:

  • In my opinion, the scientific reasoning and argumentation is shallow in several places and a few examples are given in the detailed comments below.
  • Although I consider it an interesting move to treat both growth promotion and pest control by yeasts application in the same study, it is puzzling that the former is given much more attention. For instance, the current summary in the manuscript of regulatory conditions for biostimulation products seems useful. But its inclusion appears inconsistent, since nothing is said about the fact that microorganisms for pest control are in most countries regulated within pesticide frameworks, as ‘biopesticides’ or ‘microbial pesticides’. Also, there is comparatively little information on microbiological pest/disease control in the introduction section.
  • The English in the manuscript needs substantial improvement. The text also contains quite a few repetitions and to help the reader it needs better organization, possibly with shorter paragraphs and enhanced use of sub-headings.

We really appreciate the comments provided by the reviewer that were very helpful for the improvement of our review. We agree with the reviewer that nothing was said about the fact that microorganisms for pest control are in most countries regulated within pesticide frameworks, as ‘biopesticides’ or ‘microbial pesticides’, and there is comparatively little information on microbiological pest/disease control in the introduction section. Thus, we added information about biopesticides regulations (section 2.2) and we made adjustment in our introduction section (lines 83-99) to address the points raised by the reviewer.

The English has been revised and corrected by a native. We have also sharpened focus to our key objectives and we have removed non-essential material and wording. Moreover, new sub-headings have been added throughout the review for a better organization.

A few specific comments and suggestions:

L 77-80: Unclear sentence. Yeasts may not have the same effectiveness as chemical fertilizers but of course they might have other benefits. And it is not clear which ‘problem’ is referred to here.

 We have rephrased these sentences (lines 95-99), please see the new version.

L 83-84: This sentence almost completely lacks understandable and useful content. It does not provide a good definition of yeasts, and the statement about assimilation is principally valid for all microorganisms.

Thank you for pointing this out. We have modified this phrase to be more precise with the definition of yeast (lines 100-106).

L 167-169: This sentence is extremely vague. Does it imply that since the positive effects are ‘preserved’, the microorganism needs to be applied only once? And how can something be preserved under natural conditions in in vitro conditions?

We agree that this sentence is vague and confuse. We wanted to say that microbial biostimulants activity in open field are faced with multiple/combined abiotic stresses difficult to reproduce in controlled environment. However, an effective approach would be to screen microorganisms for biostimulants activity under real field conditions and then use small-medium phenotyping platforms in controlled-environment experiments to understand their mode of action on model plants as reviewed other authors (https://doi.org/10.3389/fpls.2018.01197. Nevertheless, we have deleted this sentence from the text to avoid unnecessary information and confusion. Please see line 208 of the revised version.

L 198: It is not clear what is meant by ‘the cycle of sustainable agriculture’.

 Thank you for point it out. We have clarified this sentence (now line 237)

L 212: Lachancea is misspelled.

 We are sorry for this mistake. The genus Lachancea has been corrected (now line 252)

L 381-382: In my view the statement ‘encouraging results that consider yeast as a natural insecticide’ is scientifically problematic. First, ‘results’ cannot ‘consider’ yeasts as something. Second, the word ‘natural’ is vague and can be highly (positively) value-laden, and in my opinion should be very cautiously used in scientific texts.

We are sorry for these mistakes. We have rephrased the sentences (please see lines 424-428).

L 422-423: It is difficult to understand what the authors want to say here. Why would a better mechanistic understanding of the mechanisms of plant beneficial yeasts increase safety for consumers? And it is difficult to see a direct connection between measures to ensure a more sustainable agriculture and the safety of plant produce for human consumption.

We agree with reviewer comment. We have re-written the paragraph for a better understanding.

Table 1: The table would benefit from better organization, perhaps by grouping studies according to type of effect and/or the mechanisms.

 Thank you for the suggestion. We have re-edited Table 1 according to the type of the mechanisms. Please, see the new version in the manuscript.

L 448-474: An updated overview of commercially available organisms and products is welcome. I am not sure, however, that all these products are still available on any market. Additionally, information about in which countries the products are authorized would also be valuable.

 Thank you for your suggestions. An updated overview of commercial products have been included. We have also checked which ones are still on the market and which ones have been withdrawn. Moreover, additional information has been added focus on the type of microorganism strain used, the target microorganisms of each commercial product and the country where it was manufactured. Please, see section 6 (lines 491-514).

L 506-507: This sentence lacks scientific depth. What is meant by a yeast being ‘stable’? And what could possibly be the ‘climatic basis’ for any kind of ‘stability’ in a yeast?    

Thank you to point it out. We have corrected this sentence, now lines 563-564.

Thank you for help us to improve this review. We have corrected point by point all suggested mistakes through the manuscript. Substantial revisions have been made according to reviewer comments and recommendations. We think that thank to the reviewer comments this manuscript has been considerably improved.

Reviewer 2 Report

Line 9: I would talk of synthetic or conventional...rather than chemical...please check and modify throughout the manuscript

In the introduction, please include all the aspects you encompassed in the review. For example, I found no information about pesticide. 

Line 30: for healthy growth

Line 67: , regardless of their 

Line 71: 10–40%, which 

Line 137: biostimulants' main categories

Line 146: growth-promoting 

Line 167: the positive  

Line 171: yet, but 

Line 175: the capability of plants in the near future.

Line 205: soils, unlike 

Line 424: please include the information from table 3 https://www.mdpi.com/2304-8158/9/9/1138/htm 

Line 425: please verify if you include all the following product: Botector – BoniProtect, Aspire, Nexy, Candifruit, YieldPlus, Shemer, Noli, Sporodex; in addition, please consider the opportunity to cite the scientific literature where these products have been tested (*)

In the section on commercial formulation, please underline also the importance to consider patents (https://pubmed.ncbi.nlm.nih.gov/33550980/, https://pubmed.ncbi.nlm.nih.gov/30706832/)

Line 436: yeasts; it is

Line 455: that promote rice plants' growth

Line 466: while allowing 

Line 485: agriculture, and they 

Line 508: technique, although

(*) e.g. 

European Food Safety Authority (EFSA). Conclusion on the peer review of the pesticide risk assessment of the active substance Aureobasidium pullulans (strains DSM 14940 and DSM 14941). EFSA J 2013; 11: 3183.

Weiss A, Mögel G, Kunz S. Development of “Boni-Protect”-a yeast preparation for use in the control of postharvest diseases of apples. Proc 12th International Conference on Cultivation Technique and Phytopathological Problems in Organic Fruit-Growing. Hohenheim, Germany. 2006.Jan. 31;

del Fabbro R, Crivelli L, Lacertosa G, Digeronimo G, Calari A, Edler B, et al. Activity of Aureobasidium pullulans (Botector) against grey mold on grape, strawberry and tomato. Atti Giornate Fitopatologiche. Siena, Italy. 2016. March 8-11;

Weiss A, Weißhaupt S, Krawiec P, Kunz S. Use of Aureobasidium pullulans for resistance management in chemical control of Botrytis cinerea in berries. Acta Hortic 2014; (1017): 237-42.

Achleitner D. BOTECTOR - Effective protection against Botrytis bunch rot on grapes: influence on wine quality. Deutsche Pflanzenschutztagung 2010; 57

Wilson CL, Wisniewski ME, Droby S, Chalutz E. A selection strategy for microbial antagonists to control postharvest diseases of fruits and vegetables. Sci Hortic (Amsterdam) 1993; 53: 183-9.

Droby S, Cohen L, Daus A, Weiss B, Horev B, Chalutz E, et al. Commercial testing of Aspire: A yeast preparation for the biological control of postharvest decay of citrus. Biol Control 1998; 12: 97-101.

Sui Y, Wisniewski M, Droby S, Piombo E, Wu X, Yue J. Genome sequence, assembly, and characterization of the antagonistic yeast Candida oleophila used as a biocontrol agent against post-harvest diseases. Front Microbiol 2020; 11: 295.

El-Neshawy SM, Wilson CL. Nisin enhancement of biocontrol of postharvest diseases of apple with Candida oleophila. Postharvest Biol Technol 1997; 10: 9-14.

Bar-Shimon M, Yehuda H, Cohen L, et al. Characterization of extracellular lytic enzymes produced by the yeast biocontrol agent Candida oleophila. Curr Genet 2004; 45(3): 140-8.

Macarisin D, Droby S, Bauchan G, Wisniewski M. Superoxide anion and hydrogen peroxide in the yeast antagonist-fruit interaction: A new role for reactive oxygen species in postharvest biocontrol? Postharvest Biol Technol 2010; 58: 194-202.

Liu J, Wisniewski M, Artlip T, Sui Y, Droby S, Norelli J. The potential role of PR-8 gene of apple fruit in the mode of action of the yeast antagonist, Candida oleophila, in postharvest biocontrol of Botrytis cinerea. Postharvest Biol Technol 2013; 85: 203-9.

Wang Y, Luo Y, Sui Y, Xie Z, Liu Y, Jiang M, et al. Exposure of Candida oleophila to sublethal salt stress induces an antioxidant response and improves biocontrol efficacy. Biol Control 2018; 127: 109-15.

Lahlali R, Jijakli MH. Enhancement of the biocontrol agent Candida oleophila (strain O) survival and control efficiency under extreme conditions of water activity and relative humidity. Biol Control 2009; 51: 403-8.

European Commission (EC). Commission Implementing Regulation (EU) n°373/2013. Off. J Eur Union 2013; L112: 10-2.

Ballet N, Souche JL, Vandekerckove P. Efficacy of Candida oleophila, strain O, in preventing postharvest diseases of fruits. Acta Hortic 2016; (1144): 105-12.

Viñas I, Usall J, Teixidó N, Sanchis V. Biological control of major postharvest pathogens on apple with Candida sake. Int J Food Microbiol 1998; 40(1-2): 9-16.

Garrido CC, Usall J, Torres R, Teixidó N. Effective control of Botrytis bunch rot in commercial vineyards by large-scale application of Candida sake CPA-1. BioControl 2017; 62: 161-73.

Carbó A, Torres R, Usall J, Fons E, Teixidó N. Dry formulations of the biocontrol agent Candida sake CPA-1 using fluidised bed drying to control the main postharvest diseases on fruits. J Sci Food Agric 2017; 97(11): 3691-8.

Carbó A, Torres R, Usall J, Solsona C, Teixidó N. Fluidised-bed spray-drying formulations of Candida sake CPA-1 by adding biodegradable coatings to enhance their survival under stress conditions. Appl Microbiol Biotechnol 2017; 101(21): 7865-76.

Abadias M, Teixidó N, Usall J, Solsona C, Viñas I. Survival of the postharvest biocontrol yeast Candida sake CPA-1 after dehydration by spray-drying. Biocontrol Sci Technol 2005; 15: 835-46.

Kurtzman CP, Droby S. Metschnikowia fructicola, a new ascosporic yeast with potential for biocontrol of postharvest fruit rots. Syst Appl Microbiol 2001; 24(3): 395-9.

Karabulut OA, Tezcan H, Daus A, Cohen L, Wiess B, Droby S. Control of preharvest and postharvest fruit rot in strawberry by Metschnikowia fructicola. Biocontrol Sci Technol 2004; 14: 513-21.

Prodorutti D, Ferrari A, Pellegrini A, Pertot I. Efficacy of Metschnikowia fructicola (Shemer®) against post-harvest soft fruit (berries) rots in northern Italy (Trentino). IOBC WPRS Bull 2008; 39: 107-13.

Karabulut OA, Smilanick JL, Gabler FM, Mansour M, Droby S. Near-harvest applications of Metschnikowia fructicola, ethanol, and sodium bicarbonate to control postharvest diseases of grape in central California. Plant Dis 2003; 87(11): 1384-9.

Blachinsky D, Antonov J, Bercovitz A, Elad B, Feldman K, Husid A, et al. Commercial applications of shemer for the control of pre-and post-harvest diseases. IOBC WPRS Bull 2007; 30: 75-8.

Author Response

Responses to Reviewer 2.

Line 9: I would talk of synthetic or conventional...rather than chemical...please check and modify throughout the manuscript

We really appreciate the review comments. We have modified the word chemical throughout the manuscript, please see the new version

In the introduction, please include all the aspects you encompassed in the review. For example, I found no information about pesticide. 

Thanks to the reviewer to point it out. We have added information on pesticides and microbiological pest control. Thus, this section now refers to all the issues that are addressed in this review. Please see the new version of the manuscript.

Line 30: for healthy growth

Line 67: , regardless of their 

Line 71: 10–40%, which 

Line 137: biostimulants' main categories

Line 146: growth-promoting 

Line 167: the positive  

Line 171: yet, but 

Line 175: the capability of plants in the near future.

Line 205: soils, unlike 

We have corrected point by point all suggested mistakes through the manuscript. Please see the new attached version of the review.

Line 424: please include the information from table 3 https://www.mdpi.com/2304-8158/9/9/1138/htm 

We have added the information from Table 3 from the reference De Simone et al. 2020 as the reviewer suggested.

Line 425: please verify if you include all the following product: Botector – BoniProtect, Aspire, Nexy, Candifruit, YieldPlus, Shemer, Noli, Sporodex; in addition, please consider the opportunity to cite the scientific literature where these products have been tested (*)

Thank you for your comments. All suggested commercial products have been included. We have also checked which ones are still on the market and which ones have been withdrawn. Moreover, additional information has been added focus on the type of microorganism strain used, the target microorganisms of each commercial product and the country where it was manufactured. Please see the new version of the review.

In the section on commercial formulation, please underline also the importance to consider patents (https://pubmed.ncbi.nlm.nih.gov/33550980/, https://pubmed.ncbi.nlm.nih.gov/30706832/)

Thank you to point it out. We have added some of the suggested patents by the reviewer. Please see the new version of the manuscript.

Line 436: yeasts; it is

Line 455: that promote rice plants' growth

Line 466: while allowing 

Line 485: agriculture, and they 

Line 508: technique, although

We have corrected point by point all suggested mistakes through the manuscript. Substantial revisions have been made according to reviewer comments and recommendations. We think that thank to the reviewer comments this paper has been considerably improved.

Reviewer 3 Report

This review is aimed to give a detailed description on the potentiality of using yeasts to 106 promote sustainable agriculture, as biostimulants and biopesticide, indicating the potential mechanisms of action of yeasts in these regards. The exploitation of yeast potential as tool to confer benefits in plants and soil is the key for developing future biofertilizers, very interesting for promoting a sustainable agriculture. The review reports numerous data from very recent papers; it represents a comphrensive picture of all the aspects involved in this theme, which nowadays represents a very attractive trend. As a consequence, in the experience of the reviewer, the paper presents results of interest to the community and should be accepted for publication after minor revision.

Only minor comments have to be considered.

The figure 1 is very nice, but it contains too many information and it’s very hard to follow all the data reported in this figure. Furthermore, no correlation between schematization reported in the figure 1 and data organization in the review is present. I suggest to simplify this figure, and possibly to subdivide it in more figures.

Author Response

Responses to reviewer 3.

This review is aimed to give a detailed description on the potentiality of using yeasts to 106 promote sustainable agriculture, as biostimulants and biopesticide, indicating the potential mechanisms of action of yeasts in these regards. The exploitation of yeast potential as tool to confer benefits in plants and soil is the key for developing future biofertilizers, very interesting for promoting a sustainable agriculture. The review reports numerous data from very recent papers; it represents a comprehensive picture of all the aspects involved in this theme, which nowadays represents a very attractive trend. As a consequence, in the experience of the reviewer, the paper presents results of interest to the community and should be accepted for publication after minor revision.

Only minor comments have to be considered.

The figure 1 is very nice, but it contains too many information and it’s very hard to follow all the data reported in this figure. Furthermore, no correlation between schematization reported in the figure 1 and data organization in the review is present. I suggest to simplify this figure, and possibly to subdivide it in more figures.

We thank very much the encouraging comments. We agree with the reviewer that the figure is a little vague and too much data are reported in it. To fit better the figure to the review, we have sharpened focus to our key objectives and we have removed non-essential information. This will improve clarity of presentation and impact of the figure to the review.

Round 2

Reviewer 1 Report

Review of revision 1 of manuscript for Plants, March 2021

Hernández-Fernández, M. et al., Culturable yeasts as biofertilizers and biopesticides for a sustainable agriculture: A comprehensive reviewThe revised version of this paper has been improved in several respects. The balance between treatment of biostimulation and microbial pest/disease control is better. Some less relevant text and information has been removed. Overall, the text and the table are better organized.

However, I should be clear that my specific, mostly critical comments on the previous version were examples, and that I did note other concerns which were not included in my first review report. When these concerns are relevant for this new version, they are now included in my report on the revised version, since in my opinion they have to be taken into consideration before publication.     

I appreciate that the use of English language has been improved in some places. But the English is still not good enough for publication. This manuscript needs editing throughout by a professional in English language editing.  

Specific comments and suggestions:

L 17-20: The description in this sentence about ability to cope with various stresses can lead the reader to think that it treats stresses and stress responses in yeasts. Also, in other places treating stresses and stress responses (e.g. line 116, but have to be checked throughout), it is not sufficiently clear that the authors write about stresses to the plant. The language use and argumentation need to be improved.

L 21-24: Sentence is unclear. What is it that has a "current limited existence"? What kind of extensive research is proposed?

L 39-40: This statement would benefit from stronger referencing to literature in English.  

L 53-65: I agree that use of (at least some types of) organic fertilizers may have various negative consequences for humans or the environment. But the extremely negative picture that the authors paint of organic fertilizers, make this text sound as an argument for replacing them entirely with biostimulants. But there are also several positive aspects of using organic fertilizers and I doubt whether such fertilizers can ever be entirely abandoned. And actually, I don´t think that the author´s argument that more research is needed about microbial biostimulants and their use, need to emphasize any drawbacks of organic fertilizers in general.       

L 112-113: This definition of biocontrol/biological control strongly disagrees with definitions used in scientific literature. If the authors want to promote this definition they need to motivate that much clearer. Another alternative could be to skip the definition and simply state that antagonistic or pathogenic (e.g. entomopathogens) microorganisms are useful in biological control.

L 149-151: I am not sure that this statement is correct and suggest that the sentence is deleted. If I have understood correctly, the intention of the regulation is indeed that more microorganisms should be added to the positive list and I cannot imagine that they cannot be yeasts.  

L 152: The description of EU regulations for microbial biostimulants seems incomplete without reference to the current situation, where they are only regulated at national level. The authors might consult Traon et al. 2014 (A legal framework for plant biostimulants and agronomic fertilizer additives in the EU. Report from Arcadia International, 115 pp. Accessed 28 Nov 2019: https://orbi.uliege.be/handle/2268/169265) and Caradonia et al. 2018 (Plant biostimulant regulatory framework: Prospects in Europe and current situation at international level. Journal of Plant Growth Regulation, https://doi.org/10.1007/s00344-018-9853-4) for information.

L 182-187: The distinction between the terms "yeast" and "yeasts" need to be improved in several places throughout the manuscript. Here, the singular term is used in the heading and the text, but incorrectly, since the text actually treats yeasts as a group, and thus, plural should be used. Other examples of incorrect use of "yeast" and "yeasts" in L 537-538 and L 538-542. Please correct throughout.                  

L 214-215: I don´t understand the meaning of "... will unequivocally highlight the capability of plants in the near future". Can it be clarified?

L 269-271: Suggest that the examples of plant nutrients and in which form they are taken up are deleted. This repeats information already given in the Introduction.

L 288-290: These sentences are confusing to me, since it is not really clear that they deal with yeasts that produce plant hormones that may have effects in/on the plant. The writing can be more specific on that.

L 310: It would be nice if it could be clarified whether the authors mean that IAA synthesis pathways are characteristic but similar in all/most yeasts, or if they mean that IAA synthesis proceed by different pathways which are however characteristic for different yeasts.

L 347: The term "Dothideomycetes sp." should be "Dothideomycetes spp.", since this refers to a group of many species.

L 425-426: I would suggest the authors not to refer to this section as dealing with yeasts as "potential insecticides". The examples do not discuss insecticidal effects of yeasts, but merely biotechnical use of yeasts for production of useful substances or the use of yeasts as lures in attract-and-kill approaches.

L 447-462: I had big difficulties understanding what the message of this paragraph is, since it includes several topics and lines of reasoning. The text jumps between drawbacks of using chemical herbicides, synthetic auxin herbicides, and degradation of chemical herbicides by yeasts.

L 470-472: These sentences repeat info given earlier and can be deleted.

L 472-480: To me this discussion of Vílchez et al. appears a bit beside the point, since there are legal regulations of the utilization of domesticated microorganisms and microbial products for pest/disease control and biostimulation. These regulations contain requirements and criteria that need to be fulfilled for new strains. Of course, scientific studies like Vílchez et al. (there are more studies on this topic in the literature) are helpful to researchers assessing the safety of new strains, but in the end the regulations largely dictate which information that has to be provided.

L 481-487: In my opinion, this description of advantages is too "sunny" and positive. First, I would say that these are possible or potential advantages, since all are certainly not valid for all microbial biostimulants or pest control agents. And I disagree with point viii and suggest deletion, since "without affecting other organisms" appears an unlikely scenario in most cases.

L 500-504: I don´t think that the expression "have guaranteed environmental friendliness" is appropriate in a scientific text. It sounds more like a quote from an advertisement leaflet. Additionally, the expression "decompose potassium in the soil" is incomprehensible, since potassium is an element.

L 567-570: To say that yeasts are "the key" for development of future biofertilizers seems a strong exaggeration. I agree that they can certainly play a role and suggest that a more moderate statement would be appropriate here.

Author Response

Reviewer 1:

Hernández-Fernández, M. et al., Culturable yeasts as biofertilizers and biopesticides for a sustainable agriculture: A comprehensive reviewThe revised version of this paper has been improved in several respects. The balance between treatment of biostimulation and microbial pest/disease control is better. Some less relevant text and information has been removed. Overall, the text and the table are better organized.

However, I should be clear that my specific, mostly critical comments on the previous version were examples, and that I did note other concerns which were not included in my first review report. When these concerns are relevant for this new version, they are now included in my report on the revised version, since in my opinion they have to be taken into consideration before publication.     

I appreciate that the use of English language has been improved in some places. But the English is still not good enough for publication. This manuscript needs editing throughout by a professional in English language editing.  

We really appreciate the comments provided by the reviewer that were very helpful for the improvement of our manuscript. Substantial revisions have been made according to the new comments and recommendations throughout the text. Concerning the level of English used in this review, it was already revised by another editor and we did not have any problems on this matter. Besides, it was revised by a native speaker of English prior to our sending and we introduced all his syntactic and lexical corrections. After your comments, it has just been passed through another English correction by a native speaker and we have again introduced all his suggestions. Therefore, we hope we will not have any problems on this matter and it is now up to the standards.

Specific comments and suggestions:

L 17-20: The description in this sentence about ability to cope with various stresses can lead the reader to think that it treats stresses and stress responses in yeasts. Also, in other places treating stresses and stress responses (e.g. line 116, but have to be checked throughout), it is not sufficiently clear that the authors write about stresses to the plant. The language use and argumentation need to be improved.

We thank the reviewer for highlighting this finding. We have modified the text throughout the manuscript in order to improve the language use and argumentation.

L 21-24: Sentence is unclear. What is it that has a "current limited existence"? What kind of extensive research is proposed?

We agree with reviewer. We have clarified this sentence. Please, see lines 21-23 of the revised version.

L 39-40: This statement would benefit from stronger referencing to literature in English.  

We have added two new English references. Please, see the references 4 and 5.

L 53-65: I agree that use of (at least some types of) organic fertilizers may have various negative consequences for humans or the environment. But the extremely negative picture that the authors paint of organic fertilizers, make this text sound as an argument for replacing them entirely with biostimulants. But there are also several positive aspects of using organic fertilizers and I doubt whether such fertilizers can ever be entirely abandoned. And actually, I don´t think that the author´s argument that more research is needed about microbial biostimulants and their use, need to emphasize any drawbacks of organic fertilizers in general.       

We did not intend to indicate that organic fertilizers are extremely negative, and we have therefore altered the text to specify that organic fertilizers often need a combination with chemical fertilizers, so they still do not offer a sustainable environmental solution. Now, lines 55-58.

L 112-113: This definition of biocontrol/biological control strongly disagrees with definitions used in scientific literature. If the authors want to promote this definition they need to motivate that much clearer. Another alternative could be to skip the definition and simply state that antagonistic or pathogenic (e.g. entomopathogens) microorganisms are useful in biological control.

We apologize for this error. We have modified the sentence to clarify this. Please see lines 113-114.

L 149-151: I am not sure that this statement is correct and suggest that the sentence is deleted. If I have understood correctly, the intention of the regulation is indeed that more microorganisms should be added to the positive list and I cannot imagine that they cannot be yeasts.

L 152: The description of EU regulations for microbial biostimulants seems incomplete without reference to the current situation, where they are only regulated at national level. The authors might consult Traon et al. 2014 (A legal framework for plant biostimulants and agronomic fertilizer additives in the EU. Report from Arcadia International, 115 pp. Accessed 28 Nov 2019: https://orbi.uliege.be/handle/2268/169265) and Caradonia et al. 2018 (Plant biostimulant regulatory framework: Prospects in Europe and current situation at international level. Journal of Plant Growth Regulation, https://doi.org/10.1007/s00344-018-9853-4) for information.

We agree that this explanation is speculative at this time, and we have edited the text to state that the new Regulation (EU 2019/1009) added the most used microorganisms at this moment but yeasts were excluded. Because of this we highlighted that yeasts should be also included in the future. Please, see modifications in lines 141-153. Regarding L152, we have clarified the text on the basis of the reviewer comments and the reference (now line 141-142) https://doi.org/10.3390/molecules25051122.

L 182-187: The distinction between the terms "yeast" and "yeasts" need to be improved in several places throughout the manuscript. Here, the singular term is used in the heading and the text, but incorrectly, since the text actually treats yeasts as a group, and thus, plural should be used. Other examples of incorrect use of "yeast" and "yeasts" in L 537-538 and L 538-542. Please correct throughout.

We have corrected the terms “yeast” and “yeast” throughout the review.   

L 214-215: I don´t understand the meaning of "... will unequivocally highlight the capability of plants in the near future". Can it be clarified?

Thank you for point it out. We have clarified this sentence. Please, see the correction in the sentence in the line 215.

L 269-271: Suggest that the examples of plant nutrients and in which form they are taken up are deleted. This repeats information already given in the Introduction.

We have deleted the form in which nutrients are taken up (L269-270).

L 288-290: These sentences are confusing to me, since it is not really clear that they deal with yeasts that produce plant hormones that may have effects in/on the plant. The writing can be more specific on that.

We have corrected these sentences (L 287).

L 310: It would be nice if it could be clarified whether the authors mean that IAA synthesis pathways are characteristic but similar in all/most yeasts, or if they mean that IAA synthesis proceed by different pathways which are however characteristic for different yeasts.

We have corrected this sentence to be clearer. Please see line 309.

L 347: The term "Dothideomycetes sp." should be "Dothideomycetes spp.", since this refers to a group of many species.

We apologize for this error. The term has been corrected.

L 425-426: I would suggest the authors not to refer to this section as dealing with yeasts as "potential insecticides". The examples do not discuss insecticidal effects of yeasts, but merely biotechnical use of yeasts for production of useful substances or the use of yeasts as lures in attract-and-kill approaches.

We have corrected this query. Please, see L 424-425.

L 447-462: I had big difficulties understanding what the message of this paragraph is, since it includes several topics and lines of reasoning. The text jumps between drawbacks of using chemical herbicides, synthetic auxin herbicides, and degradation of chemical herbicides by yeasts.

We have re-edited this paragraph for a better understanding. Please see, L 447-463.

L 470-472: These sentences repeat info given earlier and can be deleted.

Thank you for point it out. We have deleted these sentences, please see the revised version of the review.

L 472-480: To me this discussion of Vílchez et al. appears a bit beside the point, since there are legal regulations of the utilization of domesticated microorganisms and microbial products for pest/disease control and biostimulation. These regulations contain requirements and criteria that need to be fulfilled for new strains. Of course, scientific studies like Vílchez et al. (there are more studies on this topic in the literature) are helpful to researchers assessing the safety of new strains, but in the end the regulations largely dictate which information that has to be provided.

We agree with reviewer. We have deleted the unnecessary information.

L 481-487: In my opinion, this description of advantages is too "sunny" and positive. First, I would say that these are possible or potential advantages, since all are certainly not valid for all microbial biostimulants or pest control agents. And I disagree with point viii and suggest deletion, since "without affecting other organisms" appears an unlikely scenario in most cases.

Several statements that we made were more ambiguous than intended, and we have adjusted the text to be clearer. Please see lines 471-478.

L 500-504: I don´t think that the expression "have guaranteed environmental friendliness" is appropriate in a scientific text. It sounds more like a quote from an advertisement leaflet. Additionally, the expression "decompose potassium in the soil" is incomprehensible, since potassium is an element.

We agree with reviewer comments. We have deleted the suggested expression and we have modified the wrong mentioned expression. Please, see L 490-491.

L 567-570: To say that yeasts are "the key" for development of future biofertilizers seems a strong exaggeration. I agree that they can certainly play a role and suggest that a more moderate statement would be appropriate here.

We agree with reviewer. We have change “key” by “basis” to be more moderate.

We thank the reviewer for help us to improve the manuscript. Substantial revisions have been made according to reviewer comments and recommendations.

Reviewer 2 Report

Please change as follows (lines 510-14): "Also the patent literature testifies the industrial interest yeasts application for the design of tailored biotechnological solutions (https://pubmed.ncbi.nlm.nih.gov/33550980/). For example, recent patents proposed specific Metschnikowia fructicola strains to improve plant performance [150] and to inhibit the growth of unwanted microorganisms [151,152]." (you have to move this part after line 532)

Author Response

We have modified the text as the reviewer suggested. Please see lines 522-525. Thank to the reviewer for help to us to improve the review.

Round 3

Reviewer 1 Report

Hernández-Fernández, M. et al., Culturable yeasts as biofertilizers and biopesticides for a sustainable agriculture: A comprehensive review

I think the authors have revised their manuscript well according to all my comments and suggestions, except for one, see below.    

Previous comment:

L 149-151: I am not sure that this statement is correct and suggest that the sentence is deleted. If I have understood correctly, the intention of the regulation is indeed that more microorganisms should be added to the positive list and I cannot imagine that they cannot be yeasts.

We agree that this explanation is speculative at this time, and we have edited the text to state that the new Regulation (EU 2019/1009) added the most used microorganisms at this moment but yeasts were excluded. Because of this we highlighted that yeasts should be also included in the future. Please, see modifications in lines 141-153. Regarding L152, we have clarified the text on the basis of the reviewer comments and the reference (now line 141-142) https://doi.org/10.3390/molecules25051122.

New comment:

Good additions of information about current regulations. But I still interpret the sentence in L 150-151 as saying that the new regulation does not regard yeasts as organisms that can possibly be added to the positive list in the future, which I don´t think is correct.